# Hybrid assembly of polymeric nanofiber network for robust and electronically conductive hydrogels

Huimin He [1,4], Hao Li[1,4], Aoyang Pu[2], Wenxiu Li[2], Kiwon Ban[2] & Lizhi Xu [1,3] ✉

Electroconductive hydrogels have been applied in implantable bioelectronics, tissue engineering platforms, soft actuators, and other emerging technologies. However, achieving high conductivity and mechanical robustness remains challenging. Here we report an approach to fabricating electroconductive hydrogels based on the hybrid assembly of polymeric nanofiber networks. In these hydrogels, conducting polymers self-organize into highly connected three dimensional nanostructures with an ultralow threshold (~1 wt%) for electrical percolation, assisted by templating effects from aramid nanofibers, to achieve high electronic conductivity and structural robustness without sacrificing porosity or water content. We show that a hydrogel composed of polypyrrole, aramid nanofibers and polyvinyl alcohol achieves conductivity of ~80 S cm$^{-1}$, mechanical strength of ~9.4 MPa and stretchability of ~36%. We show that patterned conductive nanofiber hydrogels can be used as electrodes and interconnects with favorable electrochemical impedance and charge injection capacity for electrophysiological applications. In addition, we demonstrate that cardiomyocytes cultured on soft and conductive nanofiber hydrogel substrates exhibit spontaneous and synchronous beating, suggesting opportunities for the development of advanced implantable devices and tissue engineering technologies.

Hydrogels are promising candidate materials for the construction of soft electronics and biomedical devices due to their mechanical flexibility, structural permeability, and biocompatibility. Although most traditional hydrogels are electrically inactive, recent research has begun to explore conductive hydrogels, creating new possibilities for implantable bioelectrodes[1,2], soft actuators[3,4], tissue engineering platforms[5,6], solar-powered water treatment[7,8], and other advanced technologies[9–12]. However, despite extensive research efforts, achieving high electrical conductivity and mechanical robustness in hydrogels remains challenging, which limits their practical applications[13]. For instance, hydrogels loaded with mobile ions usually exhibit a conductivity at the level of ~0.1 S cm$^{-1}$ [14]. This value is several orders of

magnitude lower than those of electronic conductors, partly because of the low mobility of ionic charge carriers[15,16]. Incorporating electronic conductors such as metal nanowires[5,17], carbon nanotubes[18,19], or conducting polymers[11,20] into hydrogels does not result in a high conductivity of the hybrid composites that are comparable to those of the fillers. This discrepancy is due to the fact that the conductive fillers are randomly distributed in hydrogel matrix, and it is difficult to achieve electrical percolation with low volume fraction (e.g., <10%) of the conductive phase[21–23]. On the other hand, increasing the concentration of the conductive fillers may compromise the mechanical properties, water content, or other attributes of the hydrogels which are essential for their functional applications.

[1]Department of Mechanical Engineering, The University of Hong Kong, Hong Kong SAR, China. [2]Department of Biomedical Sciences, City University of Hong Kong, Hong Kong SAR, China. [3]Advanced Biomedical Instrumentation Centre Limited, Hong Kong SAR, China. [4]These authors contributed equally: Huimin He and Hao Li. ✉e-mail: xulizhi@hku.hk

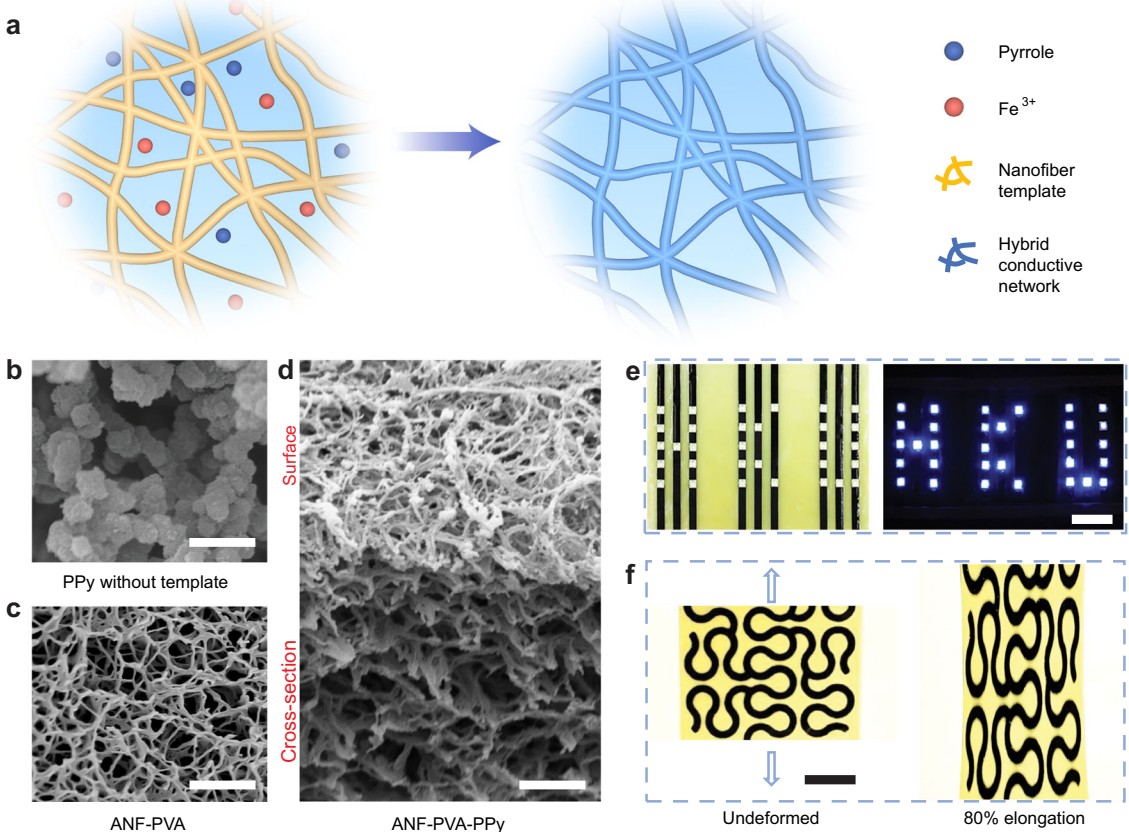

**Fig. 1 | Synthesis and structural characteristics of CNHs. a** Schematics of the synthesis of percolated polypyrrole (PPy) network guided by the nanofiber template. **b**–**d** SEM images of PPy particles polymerized in aqueous solution without nanofiber template (**b**), Aramid nanofibers-polyvinyl alcohol (ANF-PVA) template (**c**), and ANF-PVA-PPy network (**d**). Scale bars: 1 μm. **e** The electrical conductance of CNHs was demonstrated by powering an array of LEDs. Scale bars: 1 cm. **f** Photographs of serpentine conductive nanofiber hydrogel (CNH) patterns integrated on ANF-PVA substrate before (left) and after (right) 80% elongation. Scale bars: 1 cm.

Recently, organizing conducting polymers into nanoscale network was exploited for imparting high electrical conductivity to hydrogels. For example, porous structures of polyaniline (PAni) induced by freezing led to a significant increase of conductivity for hydrogels as compared with those with randomly distributed PAni particles[24]. In another scheme, aggregation of poly(3,4-ethylenedioxythiophene):polystyrene sulfonate (PEDOT:PSS) by solvent treatment and/or dry annealing resulted in hydrogels with excellent conductivity on the order of 10–47 S cm$^{-1}$[1,25,26]. However, these hydrogels usually exhibit low mechanical strengths (e.g., ~0.2 MPa) partly due to the intrinsic brittleness of the constituent conducting polymers. It was difficult to incorporate toughening components into these hydrogels without interfering with the electroconductive networks[27]. Furthermore, the dramatic volume change or high-temperature annealing involved in the materials processing raised challenges for their patterning and fabrication into hybrid devices.

Here, we report a different material system for creating robust and electronically conductive hydrogels with desired manufacturability for device applications. In this scheme, templating effect from 3D hyperconnective network of nanofibers was utilized for guiding the assembly of conducting polymers, leading to an ultralow threshold (~1 wt%) for their electrical percolation. The resultant conductive nanofiber hydrogels (CNHs) exhibit a combination of high mechanical strength and electronic conductivity originating from the hybrid polymeric nanofiber network. Furthermore, the simple processing techniques for these CNHs allow the patterning of device arrays, enabling high-performance bioelectronic tools for electrophysiological applications on cells, tissues, and organs. The outstanding electronic and mechanical properties of these CNHs, in conjunction with their manufacturability and biocompatibility, indicate new opportunities for the development of advanced soft bioelectronics and tissue engineering platforms.

## Results

### Processing and structures

Fabrication of the CNHs exploit self-assembled 3D networks involving aramid nanofibers (ANFs)[28]. Dispersing ANFs in dimethyl sulfoxide (DMSO) followed by solvent exchange with water generates hydrogels with highly connective 3D fibrillar networks, serving as templates for the assembly of conducting polymers. Incorporating polyvinyl alcohol (PVA) during the processing of hydrogels helps to weld the fibrillar joints via extensive hydrogen bonding, providing enhanced mechanical strength for the nanofiber network[29,30]. Next, monomers (e.g., pyrrole, Py) were infiltrated into the nanoporous hydrogels in an aqueous media (Fig. 1a), followed by polymerization with added oxidants (e.g., FeCl$_3$). Interestingly, this facile process led to hybrid nanofiber network with very efficient electrical percolation for the synthesized conducting polymers (i.e., polypyrrole, PPy). As indicated by scanning electron microscopy (SEM), PPy tends to form randomly distributed particles in an aqueous media without the nanofiber template (Fig. 1b). On the other hand, in the presence of ANF-PVA (Fig. 1c), the synthesized PPy conforms to the nanofiber framework with identical network topology (Fig. 1d). Successful integration of PPy on the nanofibers was also confirmed with Fourier transform infrared spectroscopy (FTIR), Raman spectroscopy, and X-ray photoelectron spectroscopy (XPS) (Supplementary Figs. 1 and 2). Indeed, the distinct

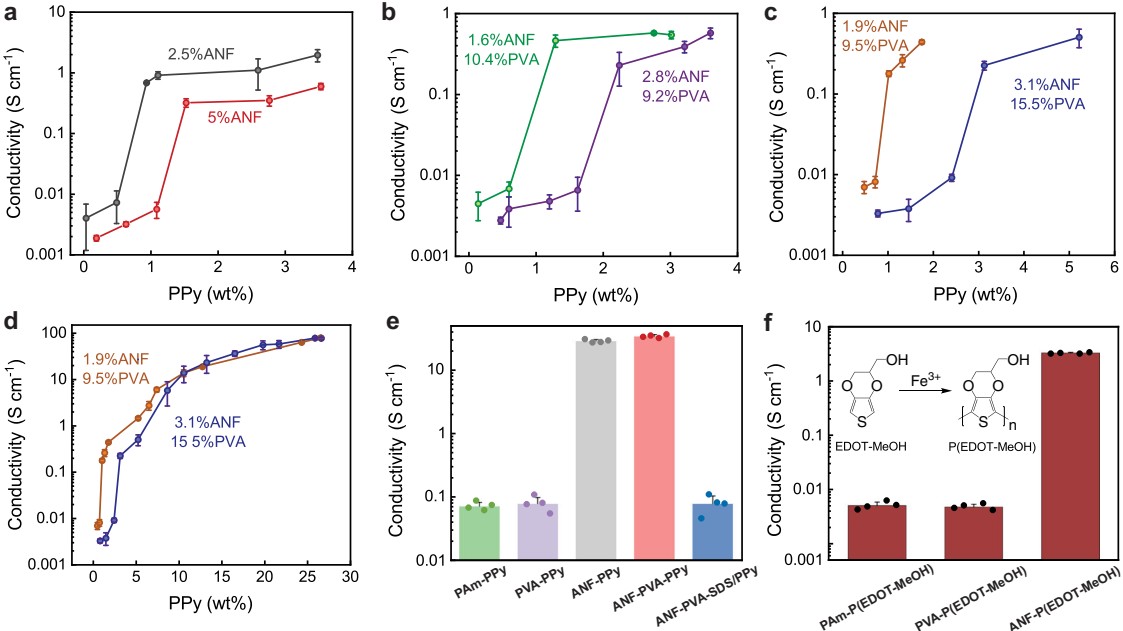

**Fig. 2 | Electrical conductivity of CNHs. a–c** Conductivity of conductive nanofiber hydrogels (CNHs) as a function of polypyrrole (PPy) content for pure aramid nanofibers (ANF) templates (**a**), ANF-polyvinyl alcohol (ANF-PVA) with a constant solid fraction of the template (**b**), and templates with a constant ratio between ANF and PVA (**c**). The electrical percolation threshold for PPy is dependent on the various configurations of the templates. **d** High conductivity of CNHs was achieved as the PPy content increased further beyond the threshold of electrical percolation. **e** Comparison of electrical conductivities between CNHs and other PPy-incorporated hydrogels prepared under similar polymerization conditions, showing the unique templating effects from ANFs. **f** Conductivity of ANF-poly (Hydroxymethylated-3,4-ethylenedioxythiophene) (P(EDOT-MeOH)) as compared with those prepared with pure PVA or polyacrylamide (PAm) matrix. Data in **a–d** and **e**, **f** are reported as their means ± SDs from $n = 3$ and $n = 4$ independent samples, respectively. Source data are provided as a Source Data file.

percolation process for PPy imparts high electrical conductance to the CNHs (Fig. 1e). The effective transport of electrons in CNHs was also indicated by its linear I-V characteristics (Supplementary Fig. 3). Furthermore, the patterning of CNHs was made possible via spatially selective synthesis of PPy in the ANF-PVA hydrogel matrix, which can generate electrodes and interconnects for soft electronic devices. The robustness of the 3D fibrillar network was translated into the macroscopic behaviors of CNHs, as exemplified by the high mechanical strength of bulk samples (Supplementary Fig. 4) and the stretchability of hybrid serpentine structures (Fig. 1f).

## Electrical behaviors

We investigated the effect of ANF networks on the electrical percolation behaviors of PPy, which provides quantitative guidance for the tuning of conductivity for CNHs. Specifically, a series of CNHs with various PPy contents were synthesized by varying the concentration of the monomer (Py) during the polymerization (Supplementary Fig. 5). The electrical conductivity of CNHs was closely related to the content of PPy, showing an abrupt increase at the threshold fraction of PPy corresponding to its electrical percolation (Fig. 2a). The conductivity of CNHs and the percolation threshold for PPy were also dependent on the various configurations of the nanofiber templates (Fig. 2a–c), providing further information about the microscopic materials assembly. To minimize artefacts induced by the non-uniformity of PPy distribution within the sample, we used hydrogel films with ~150 μm in thickness for the synthesis of PPy and the characterization of CNHs. For these thin-film samples, the concentration gradients for monomers and oxidants during the synthesis only led to a small spatial variation of the PPy content, which agreed with the results from precise characterization on sections of bulk CNH samples (Supplementary Fig. 6).

A unique feature of CNHs is that the PPy assembled into percolated conduction pathways at a very low volume fraction, which was guided by the highly connected nanofiber templates. To validate this

point, we observed a dependence of the electrical percolation threshold in CNHs on the solid content of ANFs. Specifically, the percolation threshold for PPy in CNHs with 5 wt% of ANFs is slightly above 1 wt%. In other CNHs with a lower content of ANFs (2.5 wt%), the percolation threshold for PPy dropped to the level below 1 wt% (Fig. 2a and Supplementary Fig. 7). This phenomenon can be understood as the sparser fibrillar network requires less PPy to form a continuous coating, which corresponds to a lower threshold for the formation of conduction pathway. Similar trends were also observed in various CNHs with incorporated PVA content (Fig. 2b and c). On the other hand, for the composite ANF-PVA templates, a higher amount of PPy is required to achieve electrical percolation as compared to those with pure ANFs, partly due to the higher solid content of the composite templates and the interactions between PPy and PVA. As the content of PPy raises, the conductivity of CNHs steadily increases and finally reaches an unusually high level of ~80 S cm⁻¹ (Fig. 2d) even with a high water content of ~65%. Indeed, this value was not achieved by other existing conductive hydrogels (Supplementary Fig. 8 and Table 1). Notably, the conductivity of CNH is 2-3 orders of magnitude higher than those of other PPy-based conductive hydrogels, indicating its distinct organization of conduction pathways.

The templating effect in CNHs can be attributed to the interactions between ANFs and conducting polymers. In particular, the aromatic rings and amide groups on ANFs afford attractions to PPy via π-π stacking and hydrogen bonding (Supplementary Fig. 9). Although further details of the intermolecular interactions in ANF-PVA-PPy composites are difficult to characterize, a series of control experiments demonstrated the unique templating effects arising from ANFs. Under similar conditions for the synthesis of PPy, replacing ANFs with other hydrogel matrices, such as those from pure PVA or polyacrylamide (PAm), led to a reduction of conductivity by three orders of magnitude as compared with those involving ANF templates (Fig. 2e). Microstructural examination also revealed randomly distributed PPy particles in the hydrogel matrix without ANFs, which contrasts with the

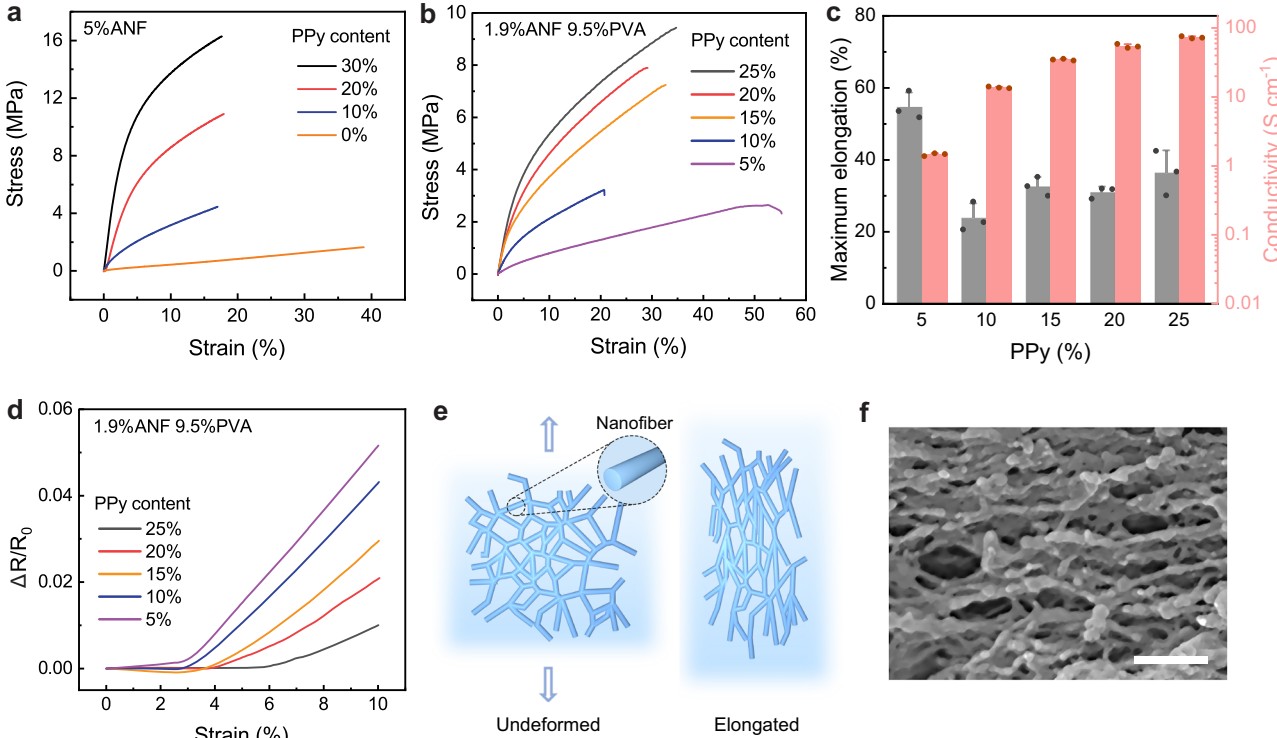

**Fig. 3 | Behaviors of CNHs under mechanical loading. a, b** Stress–strain curves of samples with various polypyrrole (PPy) content, based on pure aramid nanofibers (ANF) template (**a**) or ANF-polyvinyl alcohol (ANF-PVA) template (**b**). **c** Statistics of maximum elongation and conductivity for various conductive nanofiber hydrogels (CNHs) with 1.9% ANF and 9.5% PVA. **d** Resistance changes as a function of tensile strain for CNHs with various PPy content. **e** Schematics illustrating the orientation of the nanofiber network in response to applied strain. **f** An SEM image showing the orientation of the nanofiber network under stretching. Scale bar: 500 nm. Values in **c** represent their means ± SDs from $n = 3$ independent CNHs. Source data are provided as a Source Data file.

composite fibrillar network in CNHs (Supplementary Fig. 10). In another control experiment, adding sodium dodecyl sulfate (SDS) during the polymerization of PPy also led to a dramatic reduction in conductivity despite the existence of ANF template. This phenomenon is because the added surfactants disrupted the intermolecular interactions between ANFs and PPy, which compromised the formation of percolated conduction pathway. Furthermore, the templating effects from ANFs are also applicable to other conducting polymers such as derivatives of PEDOT. Specifically, hydroxymethyl functionalized PEDOT synthesized in the presence of ANFs led to high conductivity of the composite hydrogel, which is three orders of magnitude greater than those without templating by ANFs (Fig. 2f). On the other hand, similar templating effect for conducting polymers was also observed in other nanofiber networks such as those from bacterial cellulose (Supplementary Fig. 11), suggesting a general strategy for the synthesis of electroconductive composites.

## Mechanical properties

The hybrid assembly of polymeric nanofiber network not only imparts high electrical conductivity, but also provides structural robustness of CNHs under mechanical loading (Supplementary Fig. 12). The ANF-PPy hydrogels exhibit high strength ranging from 1.6 MPa to 17.6 MPa depending on the solid content of PPy (Fig. 3a). On the other hand, incorporation of PPy led to a decrease of failure strain for the hydrogel partly due to the intrinsic molecular rigidity of the conducting polymer. To enhance the deformability for CNHs, including flexible PVA chains in the hydrogel matrix becomes advantageous. Indeed, the ANF-PVA-PPy hydrogels exhibit a stretchability significantly higher than that of ANF-PPy (Fig. 3b and Supplementary Fig. 13). For ANF-PVA-PPy hydrogels, as the content of PPy increases from 5% to 15%, the maximum elongation firstly dropped from 55% to 20% and then increased to the level of ~34%. It is possible that incorporation of PPy

interfered with the interactions between ANF and PVA which contributed to the stretchability of the original fibrillar network. As the fraction of PPy keeps increasing, deformation of the composite hydrogel becomes more dependent on the behavior of PPy along with its interactions with ANF and PVA. In this regard, further increase of its content enhanced intermolecular interactions as well the deformability of the hybrid network. Generally, the conductivity and mechanical characteristics of CNHs can be tuned by varying the solid content (Fig. 3c and Supplementary Fig. 13-15), providing designing flexibility for functional devices.

The high robustness of CNHs is also exemplified by their strain-invariant electrical properties. Specifically, ANF-PVA-PPy samples exhibit negligible changes in resistance within a range of imposed elongation (up to ~3%-6%) (Fig. 3d and Supplementary Fig. 16a). This feature can be attributed to the re-orientation of the hybrid nanofiber network, which can accommodate macroscopic stretching without altering the topology of the conduction pathway (Fig. 3e). From SEM characterization on stretched samples, the nanofiber network exhibits significant orientation according to the mechanical load (Fig. 3f), which demonstrates the microstructural reconfiguration that contributed to the strain-invariant resistance of CNHs. As the elongation exceeds the initial regime, CNHs exhibit increases in resistance with imposed strain due to successive breakage of the conduction pathway. Nevertheless, the slopes for CNHs (0.2–0.7) are much smaller than those of typical conductive nanocomposites (e.g., 1.5–1.8 for PVA-PPy) (Supplementary Fig. 16b). Indeed, the stability of electrical conductance under deformation makes CNHs advantageous for diverse applications as stretchable electrodes and/or interconnects.

## Applications

We next explore applications of CNHs as soft bioelectronic interfaces. The electrochemical impedance per unit area for CNHs (3.1% ANF,

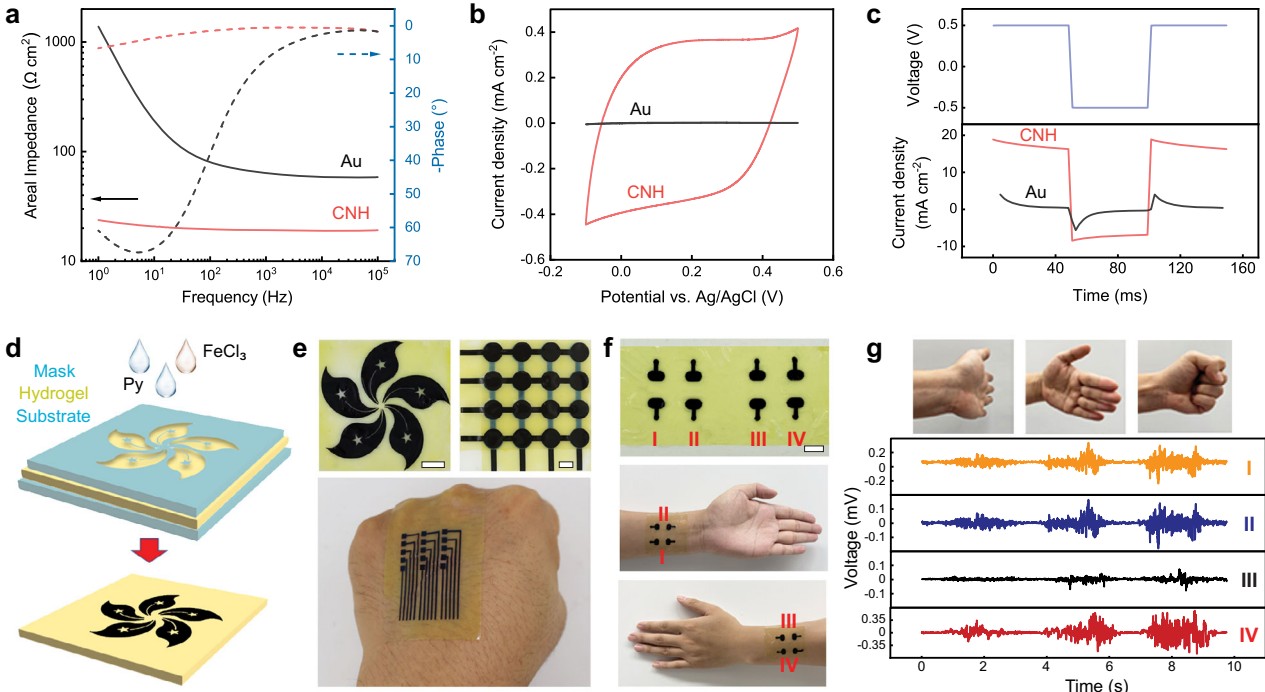

**Fig. 4 | Application of CNHs in soft bioelectronics. a, b** Electrochemical impedance spectroscopy (EIS) (**a**) and cyclic voltammetry (CV) (**b**) measurements for a conductive nanofiber hydrogel (CNH) electrode (3.1% ANF, 15.5% PVA, 25% PPy) and an Au electrode. **c** Pulsed current injection curves for a CNH electrode with the voltage switching between −0.5 and 0.5 V, as compared with responses from an Au electrode. **d** Schematic illustration of the patterning process for CNH.

**e** Photographs of patterned CNH based on ANF-PVA films. Scale bars: 1 cm. **f** Photographs of four-channel CNH electrodes mounted across the forearm. Scale bar: 1 cm. **g** Photographs of distinct hand actions and the corresponding electromyography (EMG) patterns recorded by CNH electrodes. Source data are provided as a Source Data file.

15.5% PVA, 25% PPy) in phosphate-buffered saline (PBS) is 67.2% to 98.3% lower than that of thin-film gold (Au) electrode in the physiologically relevant frequency range (Fig. 4a). At the frequency of 50 Hz, the impedance of a CNH electrode is as low as $20\,\Omega\,cm^2$, which is similar to that of PEDOT-based hydrogel electrode[1,31] and much lower than those of Au electrode ($93\,\Omega\,cm^2$) and graphene electrode ($88\,\Omega\,cm^2$) (Supplementary Fig. 17). The low impedance of CNHs originates from their extensive nanostructures providing high interfacial capacitance. This feature is beneficial for the precise characterization of various bioelectric activities with low signal amplitude. CNHs also showed high capacities in storing (Fig. 4b) and injecting (Fig. 4c) electric charges as compared with those of Au electrodes, which are desirable for delivering electrical stimulations to biological tissues. The charge injection capability (CIC) of CNH electrode is $1.2\,mC\,cm^{-2}$, which is comparable to that of PEDOT-based hydrogel electrode[25] and much higher than those of Au-based ($0.1\,mC\,cm^{-2}$) and graphene-based ($0.5\,mC\,cm^{-2}$) electrodes (Supplementary Fig. 17c). To generate millimeter-scale patterns of CNH as bioelectrodes, ANF-PVA hydrogel films (250-μm thickness) were masked with waterproof adhesive tapes and then treated with Py and $FeCl_3$ solutions (Fig. 4d). The synthesized PPy-incorporated into the ANF-PVA matrix only in the area exposed by the mask, leading to custom patterns of CNH (Fig. 4e). Monolithic hydrogel membranes involving CNH electrodes were laminated onto the skin for the measurement of electromyogram (EMG) (Fig. 4f, g) or electrocardiogram (ECG) (Supplementary Fig. 18), which captured high-quality electrophysiological signals. Distinct hand motions were characterized with EMG recordings from four independent CNH electrodes mounted on the forearm, proving a means for human-machine interactions.

Finally, CNHs showed great potential for modulating behaviors of electrogenic cells. In our pilot experiments, neonatal rat cardiomyocytes were seeded on CNH samples (3.1% ANF, 15.5% PVA, 25% PPy), as well as ANF-PVA and tissue culture plates (TCPs) for comparison. After

five days of cultivation, cardiomyocytes well spread on the surface of CNHs and TCPs, while ANF-PVA led to little attachment of cells (Supplementary Fig. 19). The improved cell adhesion on CNHs can be attributed to the decreased hydrophilicity induced by PPy (Supplementary Fig. 20), as well as interactions between its positively charged surface and negatively charged cells[32]. The cardiomyocytes grown on CNHs exhibit fast maturation indicated by good cell alignment[5,33] and pronounced expression of cardiac-specific proteins (Fig. 5a and Supplementary Fig. 19), where α-actinin and troponin T are responsible for myocardial contraction and connexin 43 is involved in gap junctions for electrical and mechanical coupling[34]. Furthermore, calcium transient imaging revealed distinct contractile behaviors of cardiomyocytes cultured on CNHs as compared to those on TCPs. Specifically, cardiomyocytes cultured on rigid and non-conductive TCPs showed weak and random excitation without coordination between cells (Fig. 5b and Supplementary Video 1). In contrast, cells attached on CNHs exhibit spontaneous and synchronous excitation with the electromechanical conduction facilitated by the soft and conductive CNH substrate (Fig. 5c and Supplementary Video 1). Coordinated contraction of cardiomyocytes on CNHs also led to macroscopic actuating the liquid media, mimicking the behavior of beating myocardium (Supplementary Video 2).

## Discussion

In summary, we have demonstrated efficient assembly of conducting polymer nanostructures templated by ANFs for the construction of electronically conductive hydrogels. The hybrid polymeric nanofiber network combines high electrical conductivity with desired mechanical robustness and manufacturability for device applications. The versatile CNHs with tunable composition and properties may expand the materials toolbox for the design of hydrogel bioelectronics, which currently focuses on quite limited options based on PEDOT[35]. We note that the processing of CNHs does not involve high temperature or

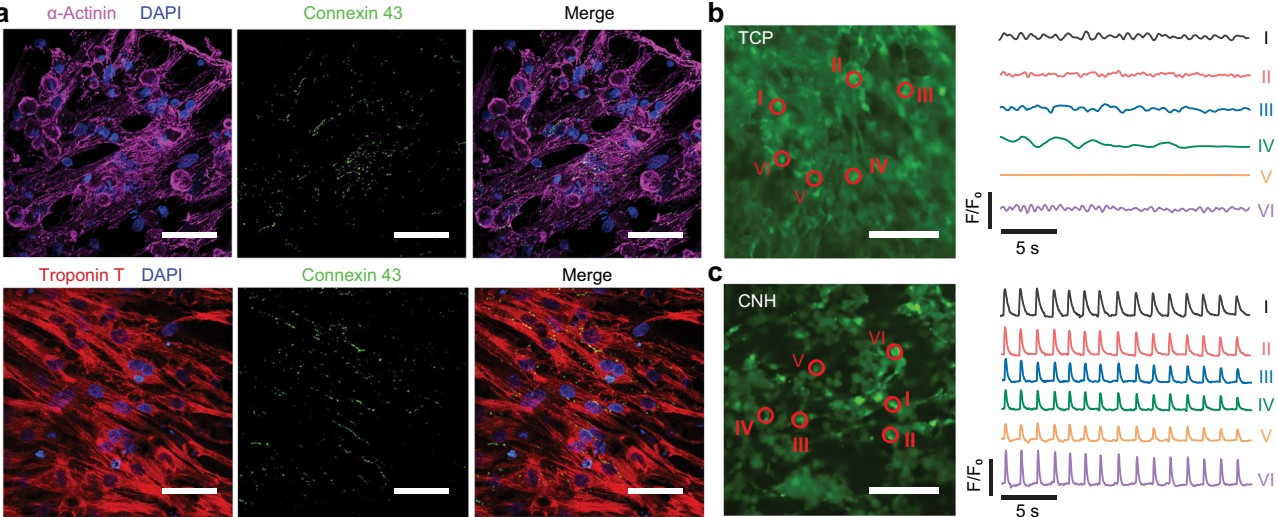

**Fig. 5 | CNHs interfaced with cardiomyocytes. a** Immunofluorescent staining of cardiac-specific proteins (α-actinin in purple; connexin 43 in green; troponin T in red) and nucleus (in blue) for neonatal rat cardiomyocytes cultured on conductive nanofiber hydrogel (CNH) (3.1% ANF, 15.5% PVA, 25% PPy). Scale bar: 50 μm. **b**, **c** Imaging of calcium transients in cardiomyocytes cultured on tissue culture plate (TCP) (**b**) and CNH (**c**), and temporal variations recorded at separate sites, showing distinct patterns of excitation. Scale bar: 100 μm. Similar results were obtained in four independent experiments. Source data are provided as a Source Data file.

dramatic volume change as those required for the reported PEDOT hydrogels[2,25]. This feature makes CNHs favorable for further integration with bioactive molecules, microfluidic channels, or other functional components for advanced bio-interfaces. The good biocompatibility of CNHs also permits the development of an implantable system for neural and cardiac applications. On the other hand, tissue engineering technologies may benefit from CNHs due to their excellent electrical properties along with biomimetic nanofibrous structures. Further tuning of mechanical behaviors, structural anisotropy, mesoscale patterning, or other attributes of CNHs may create diverse opportunities for controlling behaviors of cells and tissues through electromechanically coupled interfaces.

## Methods
### Preparation of conductive hydrogels
ANF-PVA hydrogels were fabricated according to our previously reported method[29]. Briefly, Kevlar para-aramid pulp (Type 979; DuPont) and poly (vinyl alcohol) (PVA; Mw: 146,000 − 186,000; 99%+ hydrolyzed; Sigma-Aldrich) were dissolved in dimethyl sulfoxide (DMSO; Aladdin) under magnetic stirring at 95 °C for 7 days. The resulting ANF and PVA liquids were mixed and then poured into a mold or cast on a flat steel plate using a film coater. Bulk and film hydrogels were achieved by solidification of ANF-PVA mixture through solvent exchange in deionized (DI) water for 24 h. For comparison, pure ANF, PVA, or polyacrylamide (PAm) hydrogels were also prepared. ANF hydrogels were prepared from ANF dispersion solidified by solvent exchange in DI water. PVA hydrogels were achieved using PVA solution (15% in DI water) undergoing a freeze-thaw cycle with 24 h freezing at -20 °C and thawing at room temperature. PAm hydrogels were synthesized by polymerization of acrylamide (3 M; Sigma-Aldrich) using crosslinker N,N′-methylenebisacrylamide (2 mM; Sigma-Aldrich), initiator potassium persulfate (3 mM; Sigma-Aldrich) and accelerator N,N,N′,N′-tetramethylethylenediamine (1.8 mM; Sigma-Aldrich) in deaired DI water at room temperature for 2 h. Synthesis of ANF-PVA-PPy was performed by pre-soaking ANF-PVA hydrogel in a pyrrole (Sigma-Aldrich) solution under ice-water bath with vibration at 160 rpm by a shaker. After 1 h of pre-soaking, FeCl₃ (Sigma-Aldrich) was added to the solution and the polymerization proceeded for 2 h. Samples were soaked in 0.5 mM FeCl₃ solution to preserve their high electrical conductivity. For

comparison, ANF-PPy, PVA-PPy, and PAm-PPy were fabricated using the same procedures for the polymerization of PPy. ANF-PVA-SDS/PPy was fabricated with additional SDS (0.1 M in concentration; Sigma-Aldrich) added into the aqueous solution during the polymerization. For the synthesis of PEDOT-incorporated composite hydrogels, samples were soaked in EDOT-MeOH solution (2.5 wt% in water; Sigma-Aldrich) for 1 h. FeCl₃ (1.2 times the molarity of EDOT-MeOH) was added to trigger the polymerization at room temperature for 24 h with vibration at 160 rpm by a shaker.

### Structural characterization
For SEM examination (Hitachi S4800 FEG), hydrogels were cut to expose cross-sections after plunge-freezing in liquid nitrogen, and dried by critical point drying (CPD; Tousimis Autosamdri 931). Chemical structures of samples were investigated by FTIR (Thermo Fisher IS50), Raman spectroscopy (DXRxi, Thermo Fisher), and XPS (K-Alpha, Thermo Fisher).

### Electrical and electrochemical characterization
The electrical conductance of CNHs was demonstrated by powering an array of LEDs mounted on CNH leads using silver paste. Electrical conductivity was measured by two-probe method (Keithley 2450) or a standard four-point probe (HPS2661, Helpass electronic technologies) (Supplementary Fig. 3). In two-probe method, bias voltage over the range from −0.5 to 0.5 V was applied on both ends of cuboid hydrogels with fixed length ($L$) of 25 mm, width ($W$) of 4 mm and thickness ($T$) of 0.15 mm, unless stated otherwise. The conductivity ($\kappa$, S cm⁻¹) was calculated by the following relationship:

$$\kappa = \left(\frac{I}{V}\right)\left(\frac{L}{W \times T}\right) \qquad (1)$$

where $V$ is the voltage, and $I$ is the current. Four-point probe was applied on square film samples with side length of 30 mm and thickness <0.15 mm. To study the distribution of PPy and gradient of conductivity in a thick hydrogel, an 8 mm-thick ANF-PVA-PPy sample was cut into slices of 100 μm in thickness by microtome cryostat (HM525 NX, Thermo Fisher) from the top surface downwards. The conductivity of each slice was tested and the PPy content of each slice was determined by an organic elemental analyzer (Unicube,

Elementar). Alternatively, to confirm the PPy content in the hydrogels, the conductive hydrogel was weighed first and dried completely in a vacuum oven at 80°C, followed by measuring the weight again. The solid content of the hydrogel matrix was measured before PPy loading. Then the PPy content was calculated by:

$$C_{PPy} = \frac{W_{dry} - W_{ANF-PVA}}{W_{wet}} \times 100\% \qquad (2)$$

where $C_{PPy}$ is the weight fraction of PPy in the CNH, $W_{wet}$ is the weight of the CNH in hydrated state, $W_{dry}$ is the weight of the CNH after complete dehydration, and $W_{ANF-PVA}$ is the dry weight of the hydrogel matrix. Variation of PPy content in CNHs prepared from different pyrrole concentrations was obtained by measuring six samples for every concentration. All electrical conductivity values reflect the mean of at least three samples for each condition.

For the measurement of resistance change of CNHs under applied strain, samples with a size of 25 mm × 4 mm × 0.15 mm was loaded on a tensile-compressive tester (Zwick Roell) at a deformation rate of 1% s$^{-1}$. At the meanwhile, the resistance of the conductive hydrogel was recorded until fracture.

Electrochemical impedance spectroscopy (EIS) of CNHs was recorded with a sine wave of amplitude at 10 mV. Cyclic voltammetry was performed with a scan rate of 0.15 mV s$^{-1}$. Square-wave voltage pulses were applied on CNHs from −0.5 to 0.5 V with each duration of 50 ms and the corresponding current was simultaneously recorded. All the electrochemical tests were conducted on CNHs or Au foil with exposed area of 1 cm$^2$ in PBS, using Pt foil as a counter electrode and silver/silver chloride (Ag/AgCl) as a reference electrode by an electrochemical workstation (PGSTAT302N, Metrohm Autolab).

## Mechanical tests
Mechanical tests were performed on a tensile-compressive tester (Zwick Roell). CNHs with a size of 25 mm × 4 mm × 0.15 mm were loaded with a deformation rate of 1% s$^{-1}$. For statistics of maximum elongation and conductivity of CNHs, each sample was measured for electrical conductivity first, followed by mechanical property. Reported data represents the average of three samples. Strain-stress curves were generated using origin software (orignlab 8.5) and formatting was finalized using Adobe Illustrator 2019 imaging software.

## Electrophysiological recording
Three CNH electrodes with identical geometry were used to detect ECG signals (Supplementary Fig. 18). The electrodes were adhered to a volunteer's left-forearm, right-forearm, and left leg by conductive gel (SignaGel Electrode Gel). For EMG measurement, patterned CNH electrodes arrays were attached to a volunteer's forearm, and a commercial gel electrode was attached to the elbow as a reference electrode. One end of each electrode was attached onto skin by the conductive gel, and the other end was connected to commercial data acquisition system (PowerLab T26, AD Instruments) using an alligator clip for the collection of electrophysiological signals. All the physiological experiments were standardized with the informed consent of the volunteers and approved by the Human Research Ethics Committee, The University of Hong Kong under project number EA200171.

## Cell culture
For cell experiments, CNH samples were rinsed thoroughly by PBS (Gibco) and soaked in Dulbecco's modified Eagle medium (DMEM; Gibco) supplemented with 10% fetal bovine serum (FBS; Gibco) and 1% penicillin-streptomycin (Gibco). For promoting cell attachment, samples were further treated with 0.1% gelatin solution (Type B, G1393 Sigma) under 37 °C for 1 h. CNH Neonatal rat cardiomyocytes (NRCMs) were harvested from the left ventricles of 2-day-old Sprague-Dawley rats as reported previously[36]. Briefly, ventricles were separated and collected from the neonatal hearts. After overnight digestion by 0.1% trypsin (Gibco) at 4 °C, tissues were washed with DMEM and further dissociated into single cells using 1 mg mL$^{-1}$ rat collagenase type II (Worthington) at 37 °C for 20 min. The digestion step was repeated three times until no visible tissues left following with Percoll density-gradient centrifugation for 30 min to remove non-cardiomyocyte population. After cell counting, $5 \times 10^5$ NRCMs in a volume of 20 μL were seeded onto gelatin-treated[37] CNHs or TCPs as controls. After 30 min incubation, culture medium containing 10% FBS and 1× antibiotic-antimycotic (Gibco) was added. The harvest of NRCMs from the left ventricles of 2-day-old Sprague-Dawley rats was approved by City University of Hong Kong under protocol number 11104222.

For immunocytochemistry, the cells were fixed with 4% paraformaldehyde (PFA; Santa Cruz) for 15 min at 4 °C, permeabilized with 0.2% Triton X-100 (Thermo Fisher) in PBS for another 15 min at room temperature, and blocked by 3% BSA (Sigma-Aldrich) for 1 h. Subsequently, the cells were incubated with primary antibodies to detect α-sarcomeric actinin (Anti-α-actinin, 1:100, Sigma-Aldrich A7811) or troponin T (Anti-cardiac troponin T, 1:100, Invitrogen MS-295-P1) overnight at 4 °C. After washed three times with PBS, cells were incubated for 1 h at room temperature in the dark with Alexa Fluor 488 conjugated goat anti-rabbit antibody (1:500, Invitrogen A-11008) or Alexa Fluor 568 conjugated goat anti-mouse antibody (1:500, Invitrogen A-11004). The dosage of primary and secondary antibodies is 150 μL for incubation. Cells were washed three times before adding DAPI solution (VectaShield) for nuclear staining. Samples were observed under a scanning laser confocal microscope (Eclipse Ni, Nikon).

Intracellular calcium transient imaging was performed using Fluo-4 AM (Thermo Fisher) to label calcium ions in NRCMs. Briefly, 5 days after cell seeding, hydrogels were washed twice with PBS and incubated with the calcium indicator media containing 10 μM fluo-4 AM and 0.1% Pluronic F-127 (Thermo Fisher) in PBS for 30 min at 37 °C. Next, the samples were washed with media and observed under an inverted fluorescence microscope (Eclipse Ti2, Nikon). Videos were analyzed using ImageJ2 software.

## Statistics and reproducibility
No statistical method was used to predetermine sample size. No data were excluded from the analyses. Each result presented in Figs. 1b–d, 3f, 5a–c, and Supplementary figures 10a, b, 11b, c, 19 was obtained after three independent experiments with similar results.

## Reporting summary
Further information on research design is available in the Nature Portfolio Reporting Summary linked to this article.

## Data availability
All relevant data supporting the key findings of this study are available within the article and its Supplementary Information files or from the corresponding author upon reasonable request. Source data are provided with this paper.

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

## Acknowledgements

The study is supported by Research Grants Council (RGC), and University Grants Committee (UGC) (Project 17200722 and 17200320 to L.X). The authors thank Dr. Heming Yao and professor Yoonseob Kim for fruitful discussions. The authors thank professor Chuyang Tang and Mr. Li Wang for their assistance in the surface characterization of materials.

## Author contributions

L.X. designed and supervised the research. H.H. and H.L. carried out experimental investigations. H.H., A.P., W.L., and K.B. designed and carried out cell experiments and analyses. L.X., H.H., and H.L. co-wrote the manuscript.

## Competing interests

The authors declare no competing interests.
