## [Peer Review File · Nature Communications]

REVIEWER COMMENTS

Reviewer #1 (Remarks to the Author):

In this manuscript, the authors presented an approach to prepare electronically conductive and mechanically robust hydrogels. Such outstanding electrical and mechanical properties allow the hydrogels to serve as desired materials for bioelectronic devices and tissue scaffolds. In particular, the hydrogels developed by the authors exhibit the highest conductivity among many of the recent conductive hydrogels. For hydrogels using PPy as the electroactive material, the conductivity of the authors' hydrogel is more than two orders of magnitude larger than those of others. The authors also demonstrated the applications of the hydrogels in ECG/EMG measurements and cell culture. The reviewer recommends publication of this manuscript if the authors can address the following questions and comments.

1. Mechanical properties:

The authors claimed that achieving both high conductivity and mechanical robustness in hydrogels remains challenging. However, the authors only compared the conductivity with other hydrogels. Although the authors included many testing results related to mechanical properties, they did not compare them with other hydrogels. An Ashby plot showing the conductivity and mechanical properties (e.g. failure strain, stretchability) of the authors' hydrogel and other hydrogels may be useful in this context.

Similarly, the authors mentioned that their hydrogels achieved outstanding properties without sacrificing porosity or water content. Characterizations of the porosity and water content and comparison of these properties with other hydrogels, if possible, are necessary.

2. Electrical properties:

Conductive hydrogels can be used as interconnects for electronics and electrodes for electrophysiological sensing. For the latter case, the impedance and charge injection capability (CIC), rather than the conductivity, are important metrics. The authors compared the impedance and CIC of the hydrogel with gold. Although gold is a popular material for electrode, its properties are known to be unexceptional. Can the authors compare these properties with other conductive hydrogels (e.g. PEDOT-based) and/or some other materials used for electrode (e.g. graphene, Pt black)?

3. Manufacturing:

Hydrogel is a type of ultra-soft material. How to connect the hydrogel with other rigid electronic components, such as the LEDs shown in Fig. 1e. Similarly, how to connect the hydrogel electrode in Fig. 4 to external DAQ? Adding some details in the methods section might be helpful.

It's great to see that the hydrogel can be patterned by laser machining or by a shadow mask. What are the smallest feature sizes that can be made using these two approaches? When using a shadow mask, will the solution penetrate into the edges?

4. Applications:

Can the authors specify the substrate in Fig. 4d? What is the material and thickness? In the image shown in the bottom frame of Fig. 4e, it seems that the substrate is a thin sheet of plastic/polymer that has much higher modulus compared with hydrogel. If that's the case, will the substrate increase the overall stiffness and weakens the ability of soft, conformal contact?

In another application, the author compared the cardiomyocytes cultured on TCP and hydrogel, and showed that the hydrogel can promote the maturation due to the hydrophilicity and positively charged surface. How does these properties relate to the

conductivity and robustness of the hydrogel, which are the key advantages of the authors' work. In other words, will another hydrophilic, positively-charged porous material have the same effect in promoting the maturation of cardiomyocytes? The authors need to explain why this application requires the use of the electrically conductive hydrogel.

5. Clarity:

Details of the error bars need to be added in the figure captions.

Some curves in Fig. 3 are not clear. Specifically, in Fig. 3a, the black, red and blue curves all terminate at a strain of $\sim 17\%$. Does that mean that the failure strains for all these three cases are the same? In Fig. 3b, the purple and blue curves presumably show the failure strain since there are sharp drops of the curve at $\sim 50\%$ and $\sim 20\%$, respectively. However, in the green, red and black curves, it's not clear when the hydrogels break. Some quantitative descriptions will be useful.

It will be helpful if the authors can add some quantitative information on the impedance and CIC of the hydrogel, such as the impedance at 1 kHz. It's clear to see that the impedance and CIC of the hydrogel are better than gold, but it's hard to infer the exact values from the curves in Fig. 4a and b. Readers will need these exact values to compare with other well-known materials such as PEDOT.

Reviewer #2 (Remarks to the Author):

In the manuscript entitled "Hybrid Assembly of Polymeric Nanofiber Network for Robust and Electronically Conductive Hydrogels" by Xu et al, the authors presented a novel materials system for the construction of electronically conductive hydrogels. In this scheme, conducting polymers self-assembled around aramid nanofibers to achieve efficient electrical percolation, which is quite different from the structural features of other conductive hydrogels. As a result, the hybrid 3D nanofiber network exhibits outstanding electronic conductivity and mechanical robustness, which is important for further development of complex device systems. The authors thoroughly characterized the electromechanical properties of the materials and their relationships with the processing conditions and materials composition. Methods for device fabrication were also demonstrated, leading to functional interfaces for wearable electronics and cell culture. Overall, this is a very interesting work establishing a new scheme for the engineering of advanced hydrogel bioelectronics. As demonstrated by the authors, this method may be applicable to other materials systems as well, suggesting a variety of nanofiber-templated conductive materials. Further applications of the materials in implantable devices and tissue engineering platforms are also promising. Therefore, I strongly support publication of this work in Nature Communications.

There are a few minor comments and questions for the authors' consideration:

1. In addition to the ANFs, will other nanofiber materials (such as those from cellulose) provide similar templating effects for the construction of conductive hydrogels?
2. In the main text, the authors may consider adding a few comments on the quantitative comparison between these new materials and other conductive hydrogels.
3. How did the cells attach to the CNH? Were fibronectin or other RGD-presenting ligands involved? How do they bond with the surface of CNH?

Comments from Reviewer #1

Summary: Review of “Hybrid Assembly of Polymeric Nanofiber Network for Robust and Electronically Conductive Hydrogels” by Xu et al.

In this manuscript, the authors presented an approach to prepare electronically conductive and mechanically robust hydrogels. Such outstanding electrical and mechanical properties allow the hydrogels to serve as desired materials for bioelectronic devices and tissue scaffolds. In particular, the hydrogels developed by the authors exhibit the highest conductivity among many of the recent conductive hydrogels. For hydrogels using PPy as the electroactive material, the conductivity of the authors' hydrogel is more than two orders of magnitude larger than those of others. The authors also demonstrated the applications of the hydrogels in ECG/EMG measurements and cell culture. The reviewer recommends publication of this manuscript if the authors can address the following questions and comments.

Our Response: We thank the Reviewer for the enthusiastic comments and support for the publication of our work in *Nature Communications*.

Comment #1: Mechanical properties:

The authors claimed that achieving both high conductivity and mechanical robustness in hydrogels remains challenging. However, the authors only compared the conductivity with other hydrogels. Although the authors included many testing results related to mechanical properties, they did not compare them with other hydrogels. An Ashby plot showing the conductivity and mechanical properties (e.g. failure strain, stretchability) of the authors' hydrogel and other hydrogels may be useful in this context. Similarly, the authors mentioned that their hydrogels achieved outstanding properties without sacrificing porosity or water content. Characterizations of the porosity and water content and comparison of these properties with other hydrogels, if possible, are necessary.

Our response: We are grateful to the Reviewer for the suggestion. As suggested by reviewer, we compared conductivity and mechanical strength of CNHs with other conductive hydrogels (e.g., Graphene, PANI, PEDOT and PPy) in an Ashby plot, as shown in added Supplementary Fig. 12.

Generally, in the hydrogels with nanofibers network, the pores are filled with water. Hence, there is a positive correlation between water content and porosity. Here, we characterized water content of CNHs versus PPy content in Supplementary Fig. 5c. The reduction of water content indicates the decline of porosity in CNHs due to the gradual occupation of PPy. However, we have to mention that CNHs can achieve wonderful electrical conductivity but maintaining high water content as well as porosity because of their low percolation threshold (~1 wt%). We also added some comparison of CNHs with other materials in the revised manuscript.

Our modification to the manuscript: We added Supplementary Fig. 12 in the revised manuscript on Page 8, Paragraph 2:

“The hybrid assembly of polymeric nanofiber network not only imparts high electrical conductivity, but also provides structural robustness of CNHs under mechanical loading (Supplementary Fig. 12).”

Supplementary Fig. 12 | Comparison of conductivities and mechanical strengths between CNHs and other reported conductive hydrogels. CNH exhibits outstanding electronic conductivity and mechanical strength.

On page 7, paragraph 1, we added “As the content of PPy raises, the conductivity of CNHs steadily increases and finally reaches an unusually high level of $\sim 80 \text{ S cm}^{-1}$ (Fig. 2d) even with a high water content of $\sim 65\%$.”

Comment #2: Electrical properties:

Conductive hydrogels can be used as interconnects for electronics and electrodes for electrophysiological sensing. For the latter case, the impedance and charge injection capability (CIC), rather than the conductivity, are important metrics. The authors compared the impedance and CIC of the hydrogel with gold. Although gold is a popular material for electrode, its properties are known to be unexceptional. Can the authors compare these properties with other conductive hydrogels (e.g. PEDOT-based) and/or some other materials used for electrode (e.g. graphene, Pt black)?

Our response: Many thanks to the Reviewer for this suggestion. We revised the manuscript by comparing the impedance and CIC of CNHs with PEDOT-based hydrogels and graphene as well as adding Supplementary Fig. 17 in Supplementary information.

Our modification to the manuscript: On page 9, paragraph 2, we added “At a frequency of 50 Hz, the impedance of a CNH electrode is as low as $20 \Omega \text{ cm}^2$, which is similar to that of PEDOT-based hydrogel electrode¹ and much lower than those of Au electrode ($93 \Omega \text{ cm}^2$) and graphene electrode ($88 \Omega \text{ cm}^2$) (Supplementary Fig. 17).”

On page 10, paragraph 1, we added “The charge injection capability (CIC) of CNH electrode is 1.2 mC cm^{-2} , which is comparable to that of PEDOT-based hydrogel electrode² and much higher than those of Au-based (0.1 mC cm^{-2}) and graphene-based (0.5 mC cm^{-2}) electrodes (Supplementary Fig. 17c)”

Supplementary Fig. 17 | Comparison of electrochemical properties. a, b, Electrochemical impedance spectroscopy (EIS) (a) and pulsed current injection curves for a graphene electrode with the voltage switching between -0.5 and 0.5 V vs. Ag/AgCl. (b). **c,** Cyclic electrochemical current pulse injection curves of electrodes based on CNH, Au and graphene, with the voltage switching between -0.5 V and 0.5 V vs. Ag/AgCl.

Comment #3: Manufacturing:

Hydrogel is a type of ultra-soft material. How to connect the hydrogel with other rigid electronic components, such as the LEDs shown in Fig. 1e. Similarly, how to connect the hydrogel electrode in Fig. 4 to external DAQ? Adding some details in the methods section might be helpful. It's great to see that the hydrogel can be patterned by laser machining or by a shadow mask. What are the smallest feature sizes that can be made using these two approaches? When using a shadow mask, will the solution penetrate into the edges?

Our response: As suggested by Reviewer, we have added more detailed descriptions for circuit connection in Methods section accordingly.

For the patterning, we have created CNH features at millimetre scale using laser machining or by a shadow mask. Lateral leakage of solution will reduce the precision of patterns if the feature size is at microscale. However, we found that reducing the thickness of ANF-PVA film will minimize distortion of patterns and enhance the fidelity. Precise microscale patterning could be achieved by photolithography-based technology, which is subject to future work.

Our modification to the manuscript: On page 14, paragraph 3, we added “The electrical conductance of CNHs was demonstrated by powering an array of LEDs mounted on CNH leads using silver paste.”

On page 16, paragraph 3, we added “One end of each electrode was attached onto skin by the conductive gel, and the other end was connected to commercial data acquisition system (PowerLab T26, AD Instruments) using an alligator clip for the collection of electrophysiological signals.”

On page 10, paragraph 1, we modified the sentence to “To generate millimetre-scale patterns of CNH as bioelectrodes, ANF-PVA hydrogel films (250 μm thickness) were masked with waterproof adhesive tapes and then treated with Py and FeCl₃ solutions (Fig. 4d).”

Comment #4: Applications:

Can the authors specify the substrate in Fig. 4d? What is the material and thickness? In the image shown in the bottom frame of Fig. 4e, it seems that the substrate is a thin sheet of plastic/polymer that has much higher modulus compared with hydrogel. If that's the case, will the substrate increase the overall stiffness and weakens the ability of soft, conformal contact?

In another application, the author compared the cardiomyocytes cultured on TCP and hydrogel, and showed that the hydrogel can promote the maturation due to the hydrophilicity and positively charged surface. How does these properties relate to the conductivity and robustness of the hydrogel, which are the key advantages of the authors' work. In other words, will another hydrophilic, positively-charged porous material has the same effect in promoting the maturation of cardiomyocytes? The authors need to explain why this application requires the use of the electrically conductive hydrogel.

Our response: Many thanks to the Reviewer for the comment. Actually, we generated CNH patterns on a pristine ANF-PVA film by spatially selective deposition of PPy rather than integration with another substrate. The light-yellow region of the film in Fig. 4e is ANF-PVA hydrogel without deposited PPy. More precise description has been added in the revised manuscript.

TCP and CNH can both support the attachment and maturation of cardiomyocytes. However, an electrically conductive substrate (such as CNHs) is beneficial for the functionalization of cardiomyocytes, as it facilitate transmission of electrical signals between cells, leading to coordinated contraction. Similar effects were reported in previous studies (ACS Nano 13, 163-175). We note that CNHs are advantageous for applications in cardiac tissue engineering due to their excellent electromechanical properties, manufacturability, and structural similarity to biological tissues. Further investigations on these aspects are underway and will be reported separately.

Our modification to the manuscript: On page 10, paragraph 1, we modified the sentence to “**Monolithic hydrogel membranes involving CNH electrodes were laminated onto the skin**”.

Comment #5: Clarity:

Details of the error bars need to be added in the figure captions.

Some curves in Fig. 3 are not clear. Specifically, in Fig. 3a, the black, red and blue curves all terminate at a strain of ~17%. Does that mean that the failure strains for all these three cases are the same? In Fig. 3b, the purple and blue curves presumably show the failure strain since there are sharp drops of the curve at ~50% and ~20%, respectively. However, in the green, red and black curves, it's not clear when the hydrogels break. Some quantitative descriptions will be useful.

It will be helpful if the authors can add some quantitative information on the impedance and CIC of the hydrogel, such as the impedance at 1 kHz. It's clear to see that the impedance and CIC of the hydrogel are better than gold, but it's hard to infer the exact values from the curves in Fig. 4a and b. Readers will need these exact values to compare with other well-known materials such as PEDOT.

Our response: We thank the reviewer pointing out this issue. Indeed, all the curves in figure 3a and 3b terminate at distinct failure strains. Specifically, the curves in figure 3a terminate at 17.6%, 17.9%, 17.0%, and 38.9%, respectively. In figure 3b, the failure strains were recorded at 34.7%, 29.1%, 32.7%, 20.8%, and 55.3%, respectively. For the electrochemical properties, we have added specific values of impedance and CIC of CNHs in the revised manuscript.

Our modification to the manuscript:

We added details of the statistics in the figure captions:

On Page 22, we added “Values in a-f represent mean and the error bars represent the SD of the measured values ($n \geq 3$).”

On Page 15, Paragraph 1, we added “Variation of PPy content in CNHs prepared from different pyrrole concentrations was obtained by measuring six samples for every concentration. All electrical conductivity values reflect the mean of at least three samples for each condition.”

On Page 23, we added “Values in c represent mean and the error bars represent the SD of the measured values ($n=3$).”

On Page 16, Paragraph 2, we added “For statistics of maximum elongation and conductivity of CNHs, each sample was measured for electrical conductivity first, followed by mechanical property. Reported data represents the average of three samples.”

Also, we added specific values of impedance and CIC in the revised manuscript:

On page 9, paragraph 2, we added “At the frequency of 50 Hz, the impedance of a CNH electrode is as low as $20 \Omega \text{ cm}^2$, which is similar to that of PEDOT-based hydrogel electrode¹ and much lower than those of Au electrode ($93 \Omega \text{ cm}^2$) and graphene electrode ($88 \Omega \text{ cm}^2$) (Supplementary Fig. 17).”

On page 10, paragraph 1, we added “The charge injection capability (CIC) of CNH electrode is 1.2 mC cm^{-2} , which is comparable to that of PEDOT-based hydrogel electrode² and much higher than those of Au-based (0.1 mC cm^{-2}) and graphene-based electrodes (0.5 mC cm^{-2}) (Supplementary Fig. 17c)”

Comments from Reviewer #2

Summary: In the manuscript entitled “Hybrid Assembly of Polymeric Nanofiber Network for Robust and Electronically Conductive Hydrogels” by Xu et al, the authors presented a novel materials system for the construction of electronically conductive hydrogels. In this scheme, conducting polymers self-assembled around aramid nanofibers to achieve efficient electrical percolation, which is quite different from the structural features of other conductive hydrogels. As a result, the hybrid 3D nanofiber network exhibits outstanding electronic conductivity and mechanical robustness, which is important for further development of complex device systems. The authors thoroughly characterized the electromechanical properties of the materials and their relationships with the processing conditions and materials composition. Methods for device fabrication were also demonstrated, leading to functional interfaces for wearable electronics and cell culture. Overall, this is a very interesting work establishing a new scheme for the engineering of advanced hydrogel bioelectronics. As demonstrated by the authors, this method may be applicable to other materials systems as well, suggesting a variety of nanofiber-templated conductive materials. Further applications of the materials in implantable devices and tissue engineering platforms are also promising. Therefore, I strongly support publication of this work in *Nature Communications*.

There are a few minor comments and questions for the authors’ consideration:

Our Response: We thank the Reviewer for the comments and recommendation of publishing our work in *Nature Communications*.

Comment #1: In addition to the ANFs, will other nanofiber materials (such as those from cellulose) provide similar templating effects for the construction of conductive hydrogels?

Our Response: We thank the reviewer for the point raised. To address this comment, bacterial cellulose (BC) hydrogels were used to test the templating effects. The same method of fabricating ANF-PPy hydrogels was utilized to synthesize BC-PPy hydrogels. We found that the phenomenon shown in ANF-PPy hydrogels also exist in BC-PPy hydrogels. As shown in the Supplementary Fig. 11 appended below, the conductivity of BC-PPy hydrogels is two orders of magnitude higher than that of BC-PPy-SDS samples. The SEM images of BC and BC-PPy samples both demonstrate fibrous structures.

Our modification to the manuscript: We added Supplementary Fig.11 and a brief discussion as “On the other hand, similar templating effect for conducting polymers was also observed in other nanofiber networks such as those from bacterial cellulose (Supplementary Fig. 11), suggesting a general strategy for the synthesis of electroconductive composites.” in the revised manuscript on Page 8, Paragraph 1.

Supplementary Fig. 11 | Templating effect of bacterial cellulose (BC) hydrogels. a, Comparison of electrical conductivities between BC-PPy and BC-PPy-SDS hydrogels prepared under similar polymerization conditions. **b-c,** SEM images of BC (**b**) and BC-PPy (**c**). The PPy polymerized along the nanofibers indicates the templating effect of BCs. Scale bars: 1 μm.

Comment #2: In the main text, the authors may consider adding a few comments on the quantitative comparison between these new materials and other conductive hydrogels.

Our response: Following the reviewer’s suggestion, we have quantitatively compared CNHs with other reported hydrogels with more comments in the revised manuscript.

Our modification to the manuscript:

We added quantitative comparison between CNHs and other hydrogels as below:

On page 7, paragraph 1, we added “Notably, the conductivity of CNH is 2-3 orders of magnitude higher than those of other PPy-based conductive hydrogels, indicating its distinct organization of conduction pathways.”

On page 9, paragraph 2, we added “At the frequency of 50 Hz, the impedance of a CNH electrode is as low as 20 Ω cm², which is similar to that of PEDOT-based hydrogel electrode¹ and much lower than those of Au electrode (93 Ω cm²) and graphene electrode (88 Ω cm²) (Supplementary Fig. 17).”

On page 10, paragraph 1, we added “The charge injection capability (CIC) of CNH electrode is 1.2 mC cm⁻², which is comparable to that of PEDOT-based hydrogel electrode² and much higher than those of Au-based (0.1 mC cm⁻²) and graphene-based electrodes (0.5 mC cm⁻²) (Supplementary Fig. 17c)”

Comment #3: How did the cells attach to the CNH? Were fibronectin or other RGD-presenting ligands involved? How do they bond with the surface of CNH?

Our response: In this study, 0.1% gelatin was used to treat the samples by incubated at 37 °C for 1h. We have added more details in the modified manuscript.

Our modification to the manuscript: On page 20, paragraph 4, we added “**For promoting cell attachment, samples were further treated with 0.1% gelatin solution (Type B, G1393 Sigma) under 37°C for 1h.**”

1. Jiang Y, *et al.* Topological supramolecular network enabled high-conductivity, stretchable organic bioelectronics. *Science* **375**, 1411-1417 (2022).
2. Lu B, *et al.* Pure PEDOT:PSS hydrogels. *Nature Communications* **10**, 1043 (2019).

REVIEWERS' COMMENTS

Reviewer #2 (Remarks to the Author):

The revised manuscript has addressed my comments, which is recommended for publication.

Comments from Reviewer #2

The revised manuscript has addressed my comments, which is recommended for publication.

Our Response: We thank the Reviewer support for the publication of our work in *Nature Communications*.